# Lidar-Based Navigation of Subterranean Environments Using Bio-Inspired Wide-Field Integration of Nearness

**DOI:** 10.3390/s22030849

**Published:** 2022-01-23

**Authors:** Michael T. Ohradzansky, J. Sean Humbert

**Affiliations:** 1Department of Aerospace Engineering Sciences, University of Colorado Boulder, 3775 Discovery Drive, Boulder, CO 80303, USA; 2Department of Mechanical Engineering, University of Colorado Boulder, 427 UCB, 1111 Engineering Dr, Boulder, CO 80309, USA; sean.humbert@colorado.edu

**Keywords:** bio-inspired navigation, sensorimotor convergence, subterranean exploration, quadrotor control

## Abstract

Navigating unknown environments is an ongoing challenge in robotics. Processing large amounts of sensor data to maintain localization, maps of the environment, and sensible paths can result in high compute loads and lower maximum vehicle speeds. This paper presents a bio-inspired algorithm for efficiently processing depth measurements to achieve fast navigation of unknown subterranean environments. Animals developed efficient sensorimotor convergence approaches, allowing for rapid processing of large numbers of spatially distributed measurements into signals relevant for different behavioral responses necessary to their survival. Using a spatial inner-product to model this sensorimotor convergence principle, environmentally relative states critical to navigation are extracted from spatially distributed depth measurements using derived weighting functions. These states are then applied as feedback to control a simulated quadrotor platform, enabling autonomous navigation in subterranean environments. The resulting outer-loop velocity controller is demonstrated in both a generalized subterranean environment, represented by an infinite cylinder, and nongeneralized environments like tunnels and caves.

## 1. Introduction

There is significant interest in using small Unmanned Aerial Systems (sUAS) for autonomous tasks, ranging from rapid situational awareness of hazardous environments to search and rescue missions. In each application, the system’s ability to sense the environment around itself is critical to autonomous navigation, and generating control commands in a fast and efficient manner is still an ongoing challenge. This challenge is even more difficult in subterranean environments, where degraded sensing, diverse unstructured terrain, and a lack of Global Positioning System (GPS) signals can cause problems with traditional navigation methods. More detail on the challenges and proposed solutions to autonomous subterranean navigation are presented in [1,2,3,4,5,6,7,8,9,10,11,12]. Generally, proposed methods for subterranean navigation include some mix of the following subsystems: sensing and perception, state-estimation, map generation, planning, and guidance. These subsystems and processes can require multiple sensing modalities and significant computing resources, which ultimately leads to undesired size, weight, and power (SWaP) constraints on the vehicle. Additionally, all of these processes are directly affected by state-estimation accuracy; if the state estimates are noisy or experience significant drift over time, it can affect the quality of the map, which in turn affects the system’s ability to navigate. To overcome these challenges and limitations, novel sensors and sensor processing methods need to be developed to continue to push forward robot autonomy.

For inspiration, researchers can look to countless examples in nature of elegant solutions to complex perception and navigation problems. Animals evolved for hundreds of millions of years. As a result, they developed efficient sensorimotor systems for processing spatially distributed sensor measurements. As engineers, we can learn from these incredible biological systems and use the underlying principals to come up with unique solutions to the different challenges of robot autonomy.

One example is the mechanosensory lateral line system, which allows fish to navigate their surroundings by sensing water flow. The mechanosensory lateral line system is composed of two different types of spatially distributed sensors: the superficial neuromast (SN) and the canal neuromaster (CN). The SNs are located on the body of the fish and sense local flow velocity, while the CNs are located in subepidermal canals and sense flow acceleration [13,14]. Specialized neural pathways enable the fish to achieve different behaviors like rheotaxis [15,16], schooling [17], obstacle avoidance [13], and prey detection [18,19]. By mimicking the mechanosensory lateral line system, reflexive obstacle avoidance can be achieved through the spatial decomposition of electric field measurements [20,21]. Another example is the spider mechanosensory system composed of hair and strain sensors that cover the abdomen and legs [22]. By processing many different spatially distributed measurements from the touch sensitive hairs, spiders are more equipped to detect and localize prey [23,24]. These hairs also allow them to sense air flow, allowing them to localize airborne prey and estimate ego-motion [25,26,27]. A third example of particular interest is the insect visuomotor system, which enables fast flight and obstacle avoidance. The insect compound eye is composed of thousands of ommatidea, or light receptors. Insects rely on optic flow, or the characteristic patterns of visual motion that are excited as they move [28,29,30]. By processing the changing patterns of light (optic flow) sensed by the ommatidia (the insect compound eye), insects are able to generate reactive motor commands allowing them to make rapid adjustments to their flight and avoid obstacles [31]. Additional analysis of the insect visuomotor system shows how specialized tangential cells process distributed optic flow measurements, inspiring the wide-field integration framework presented in [32,33,34]. Through observation of insects and other flying animals in different environments, optic flow-based navigation methods were developed [35,36,37,38,39,40,41,42]. Small-obstacle avoidance algorithms based on optic flow measurements are demonstrated onboard a multirotor platform in [43,44].

In each of these examples, hundreds to thousands of spatially distributed sensor measurements are processed simultaneously to produce different behavioral responses. Past studies focused on extracting ego-motion estimates and/or relative proximity to obstacles from optic flow or electric field measurements, whereas the main sensing mode for this study is relative depth observed by a spherical LiDAR sensor model. The proposed controller is reactive, which means that it generates control commands directly from the raw sensor measurements. Other reactive control approaches can be divided into the following subgroups: artificial potential fields, admissable-gap navigation, and laser-based reactive control approaches. In artificial potential field approaches, the known, or sensed, environment is converted to a field of forces [45,46,47,48]. The admissible-gap approach uses depth scans to generate a nearness diagram and then identify gaps to traverse [49]. The approach is strictly planar, however improvements to the initial algorithm were made including smooth nearness-diagram navigation [50], closest gap navigation [51], and tangential gap navigation [52]. This approach was recently implemented onboard a small multirotor platform and tested in both indoor and outdoor environments [53]. Other noteworthy LiDAR-based approachs are presented in [54], where depth scans are used to plan a path through the known environment using an improved rapidly-exploring random tree algorithm (RRT*) and [8], where depth scans are used to generate a centering response for graph navigation.

In this work, a bioinspired wide-field integration (WFI) framework for subterranean navigation is presented where the primary sensing modality is measured depth. Spatially distributed depth measurements are a rich source of information relevant to navigation. Using the spatial inner-product, a WFI method inspired by the Lobula Plate Tangential Cells (LPTCs) of the insect visuomotor system, environment relative states can be extracted from depth scans and applied as feedback to achieve a 3D centering response in subterranean environments. With careful selection of 3D weighting shape functions, or sensitivity shapes, control commands can be computed directly from depth scans in a single operation. The control algorithm is demonstrated onboard a simulated quadrotor platform in both a generalized cylinder environment and other subterranean environments with more diverse topology. The main advantage to this approach is its simplicity: it does not require state-estimates, local or global maps, or planned paths, and control commands are generated directly from each successive depth scan. By integrating across thousands of depth measurements, the algorithm is robust to sensor noise. The algorithm is also robust to changes in the environment. In the development of the algorithm the local environment is approximated as an infinite cylinder, however, navigation in more diverse subterranean environments is demonstrated.

This paper is outlined as follows. In Section 2, the development of the bio-inspired nearness controller is presented, as well as a description of the platform and simulated testing environment. Following the presentation of the algorithm, results from autonomous flight tests are presented in Section 3. The advantages and limitations of the controller are discussed in Section 4.

## 2. Materials and Methods

### 2.1. System and Sensing Models

For the purpose of testing the algorithm, the DARPA SubT Simulator is used (Available online: https://www.subtchallenge.com (accessed on 17 January 2022). The DARPA SubT Simulator provides access to a number of different simulated subterranean environments that fall into three main subdomains: tunnel, urban, and cave. Each subdomain provides unique navigation challenges to autonomous platforms, such as the tight and narrow corridors of the tunnel worlds or the nonuniform caverns present in the cave worlds. Examples of these environments are shown in Figure 1. Initial formulation of the controller assumes an infinite cylinder as the generalized environment, which can also be loaded into the DARPA SubT simulator as a custom world. The SubT virtual code repository and resources on how to use it are hosted on GitHub (Available online: https://github.com/osrf/subt/wiki (accessed on 17 January 2022).

In addition to simulation worlds, the DARPA SubT Simulator provides access to robotic platforms developed by DARPA SubT Challenge Virtual competition teams complete with full sensor suites. Users can customize different platforms, like multirotor aerial systems or quadrupedal ground vehicles, with sensors that suite the needs of their algorithms. For the purpose of this study, the standard quadrotor “X3” model is used with outer-loop velocity control. Navigation of the different environments is achieved by setting a fixed forward speed and generating lateral and vertical velocity commands as well as heading rate commands. The inner-loop attitude controller is handled by the DARPA SubT Simulator quadrotor controller, which converts velocity commands to individual motor speeds of the simulated vehicle for attitude control and stabilization. A depth sensing model based on existing LiDAR sensors is used to measure depth to discrete points in the environment. Current LiDAR-based sensors have limited field of views, however for the purpose of this study a LiDAR sensor with a full spherical field of view is used. Depth measurements $d(\theta_{v},\phi_{v},x)$ are discretized according to the body-referred viewing angles $\theta_{v}$ and $\phi_{v}$, where $\theta_{v}\in \left[0,2\pi \right]$ is the inclination angle and $\phi_{v}\in \left[-\pi ,\pi \right]$ is the azimuthal angle, and are a function of the environment relative system states x, which are described in greater detail in Section 2.2.1. In this case, 64 equally spaced inclination angles and 360 equally spaced azimuthal angles are used, resulting in just over 23,000 depth measurements per scan. In the Robot Operating System (ROS), the incoming depth scans are represented as collections of (x,y,z) points, and so each depth measurement is made up of 3 floating point numbers. Depth scans are reported at a rate of 20 Hz, and the bandwidth of the sensor data is estimated to be around 5.27 MB/s. The LiDAR model has a max sensing distance $d_{max}=100$ m and minimum sensing distance $d_{min}=0.05$ m. Additive Gaussian white noise is added to each depth measurement, with a nominal standard deviation of 0.03. Measured depth is hypothetically unbounded, and so depth is converted to nearness, as shown in Equation (Equation 1):(1)$$\mu \left(\theta_{v},\phi_{v},x\right)=\frac{1}{d(\theta_{v},\phi_{v})}\in \left(0,d_{min}^{-1}\right)$$

### 2.2. Bio-Inspired Nearness Control

The goal is to develop an outer-loop velocity controller that produces a 3D centering response in subterranean environments using spatially distributed nearness measurements, or nearness scans. This is achieved by extracting environment relative information from the nearness scan using the spatial inner-product. First, the local environment is approximated as an infinite cylinder, which appropriately represents a subterranean environment where large obstacles can exist above, below, and to the sides of the vehicle. When navigating unknown environments, it is often desirable to fly as far away from large obstacles, such as walls, floors, and ceilings, as possible. In the case of subterranean environments, this can be achieved by driving the system to the middle of the local environment. By selecting appropriate weighting shapes for the spatial inner-product, these environment relative states can be extracted from the nearness scans of the onboard sensor. Linear feedback control is used to drive the system away from obstacles towards the center of the cylinder and point down the centerline. The general process is shown in Figure 2.

Section 2.2.1 presents the lytic function for nearness in the approximated local environment. In Section 2.2.2, the process of extracting the environment relative states through WFI is presented in detail. Design of the linear feedback controller is presented in Section 2.2.3.

#### 2.2.1. Parameterization of the Environment

It is useful to approximate the local environment around the vehicle as some generalized shape with a known analytic representation. This enables the determination of the environment relative states to use for feedback and also allows an initial set of feedback gains to be derived. As stated above, a cylinder is selected as the generalized local environment, as it simulates the presence of obstacles above, below, and to the sides of the vehicle. Figure 3 shows different views of a quadrotor model flying through a simple cylinder environment. The controller goal is to drive the system towards the centerline of the cylinder, and the environment relative states that can be used for feedback to achieve this are the lateral, vertical, and heading displacements from the cylinder centerline defined as *y*, *z*, and ψ. As the system moves through the environment, the expected shape of the measured nearness can be determined by intersecting a line in the body-referred spherical coordinate system with the surface of a cylinder (see Appendix A for a more detailed derivation). One simplifying assumption is made, which is that the roll and pitch angles of the vehicle are assumed to be negligibly small. The analytic representation of nearness as a function of the cylinder radius *r*, the body-referred viewing angles ($\theta_{v}$, $\phi_{v}$), and the environment relative states $x={\left[\begin{array}{ccc} y & z & \psi \end{array}\right]}^{T}$ is shown in the following equation:(2)$$\mu \left(\theta_{v},\phi_{v},x\right)=\frac{c{\left(\theta_{v}\right)}^{2}+s{\left(\phi_{v}+\psi \right)}^{2}s{\left(\theta_{v}\right)}^{2}}{zc\left(\theta_{v}\right)+ys\left(\phi_{v}+\psi \right)s\left(\theta_{v}\right)+\sqrt{ac{\left(\theta_{v}\right)}^{2}+s\left(\phi_{v}+\psi \right)[bs\left(\phi_{v}+\psi \right)s{\left(\theta_{v}\right)}^{2}+yzs\left(2\theta_{v}\right)]}}$$
where sin(x) and cos(x) were simplified to s(x) and c(x), a=(r−y)(r+y), and b=(r−z)(r+z). While this is a complicated nonlinear function, estimates of the environment relative states can be extracted using WFI.

#### 2.2.2. Wide-Field Integration

In the visuomotor system of the fly, tangential cells are large motion-sensitive neurons that are sensitive to different flow patterns. Different tangential cells are sensitive to different stimulus patterns, and the integrated output is effectively a comparison between the cell’s sensitivity pattern and the measured stimulus. The outputs of these cells are pooled to produce different motor responses. This process can be represented mathematically by the spatial inner-product, shown in Equation (Equation 3), which compares two spherical nearness patterns.
(3)$$\langle \mu ,F_{i}\rangle ={\;\int \int }_{\Omega }\mu \left(\theta_{v},\phi_{v},x\right)F_{i}\left(\theta_{v},\phi_{v}\right)d\Omega$$

Here, $d\Omega =\sin\left(\theta_{v}\right)d\theta_{v}d\phi_{v}$ is the solid angle of the sphere. By projecting nearness onto different weighting shape functions $F_{i}$, the function’s dependence on the viewing angles $\theta_{v}$, and $\phi_{v}$ is removed. Real spherical harmonics, shown in Figure 4, are used as the basis set for the sensitivity shapes, and are parameterized by the viewing angles $\theta_{v}$ and $\phi_{v}$ according to:(4)$$Y_{l}^{m}\left(\theta_{v},\phi_{v}\right)=\sqrt{\frac{\left(2l+1\right)}{4\pi }\frac{(l-m)!}{(l+m)!}}P_{l}^{m}\left(\cos\theta_{v}\right)e^{im\phi_{v}}$$
where $P_{l}^{m}\left(\cos\theta \right)$ is the associated Legendre function l∈Z, m∈Z, l≥0, |m|≤l. With spherical harmonics as the weighting shapes $F_{i}$, Equation (Equation 3) can be rewritten as:(5)$$p_{i}\left(x\right)=\langle \mu ,Y_{l}^{m}\rangle ={\;\int \int }_{\Omega }\mu \left(\theta_{v},\phi_{v},x\right)Y_{l}^{m}\left(\theta_{v},\phi_{v}\right)\sin\left(\theta_{v}\right)d\theta_{v}d\phi_{v}$$

The resulting outputs $p_{i}\left(x\right)$ from Equation (Equation 5) are nonlinear functions of the state, and can be linearized for small motions around $x_{0}$ to produce a set of linear equations of the form p=Cx. This observation model can be inverted to produce a linear relationship between the projections and the environment relative states $x=C^{†}p$, where $C^{†}={\left(C^{T}C\right)}^{-1}C^{T}$. The $C^{†}$ matrix relating the projections to the states for the first nine spherical harmonics is found to be:(6)$$C^{†}=\left[\begin{array}{ccccccccc} 0 & 0 & 0 & \frac{r^{2}}{2\sqrt{\frac{\pi }{3}}} & 0 & 0 & 0 & 0 & 0\\ 0 & \frac{r^{2}}{2\sqrt{\frac{\pi }{3}}} & 0 & 0 & 0 & 0 & 0 & 0 & 0\\ 0 & 0 & 0 & 0 & 0 & 0 & 0 & 0 & \frac{3r^{2}}{8\sqrt{\frac{\pi }{3}}} \end{array}\right]$$

The spatial inner-product is a linear operator, and so the $C^{†}$ matrix, which relates projections to states, can be moved inside the inner-product resulting in the following:(7)$${\hat{x}}_{i}=\langle \mu ,\sum_{j=1}^{M}C_{ij}^{†}F_{j}\rangle$$
where *M* is the number of weighting shapes. The second term in the inner-product can be interpreted as the weighting shape for extracting the *i*’th environment relative state:(8)$$F_{{\hat{x}}_{i}}=\sum_{j=1}^{M}C_{ij}^{†}F_{j}$$

For the first nine spherical harmonics, a single projection correlates with each of the three environment relative states. The closed forms of the state sensitivity shapes are shown in Equations (Equation 9) and (Equation 10), and a visual representation is shown in Figure 5.
(9)$$F_{\hat{y}}=\frac{r^{2}}{2\sqrt{\frac{\pi }{3}}}Y_{1}^{-1}\left(\theta_{v},\phi_{v}\right)$$
(10)$$F_{\hat{z}}=\frac{r^{2}}{2\sqrt{\frac{\pi }{3}}}Y_{0}^{1}\left(\theta_{v},\phi_{v}\right)$$
(11)$$F_{\hat{\psi }}=\frac{3r^{2}}{8\sqrt{\frac{\pi }{3}}}Y_{2}^{-2}\left(\theta_{v},\phi_{v}\right)$$

#### 2.2.3. Feedback Control Design

Quadrotor platforms are able to move in any direction, subject to dynamic constraints. At low speeds, the linear dynamics models for the forward, lateral, vertical, and heading states can be decoupled and represented by the following dynamics models:(12)$$\dot{u}=X_{u}u+X_{\delta_{Fwd}}\delta_{Fwd}$$
(13)$$\left[\begin{array}{c} \dot{y}\\ \dot{v} \end{array}\right]=\left[\begin{array}{cc} 0 & 1\\ 0 & Y_{v} \end{array}\right]\left[\begin{array}{c} y\\ v \end{array}\right]+\left[\begin{array}{c} 0\\ Y_{\delta_{Lat}} \end{array}\right]\delta_{Lat}$$
(14)$$\left[\begin{array}{c} \dot{z}\\ \dot{w} \end{array}\right]=\left[\begin{array}{cc} 0 & 1\\ 0 & Z_{w} \end{array}\right]\left[\begin{array}{c} z\\ w \end{array}\right]+\left[\begin{array}{c} 0\\ Z_{\delta_{Vert}} \end{array}\right]\delta_{Vert}$$
(15)$$\left[\begin{array}{c} \dot{\psi }\\ \dot{r} \end{array}\right]=\left[\begin{array}{cc} 0 & 1\\ 0 & N_{r} \end{array}\right]\left[\begin{array}{c} \psi \\ r \end{array}\right]+\left[\begin{array}{c} 0\\ N_{\delta_{Head}} \end{array}\right]\delta_{Head}$$

Equation (Equation 12) shows the forward velocity dynamics of the vehicle where $X_{u}$ is the forward, or longitudinal, stability derivative and $X_{\delta_{Fwd}}$ is the forward control derivative. A fixed, or regulated forward speed is desired, and no forward position states are required. Equation (Equation 13) shows the lateral dynamics model of the system where $Y_{v}$, $Y_{p}$, $L_{v}$, and $L_{p}$ are the lateral stability derivatives and $Y_{\delta_{Lat}}$ and $L_{\delta_{Lat}}$ are the lateral control derivatives. Similarly, Equation (Equation 14) shows the vertical dynamics model where $Z_{w}$ is the vertical stability derivative and $Z_{\delta_{Vert}}$ is the vertical control derivative. Last is Equation (Equation 15), which shows the heading dynamics of the system where $N_{r}$ is the heading stability derivative and $N_{\delta_{Head}}$ is the heading control derivative. The inputs to the system are the forward, lateral, vertical, and heading control efforts $\delta_{Fwd}$, $\delta_{Lat}$, $\delta_{Vert}$, and $\delta_{Head}$. The table in Appendix B lists the values used for all the stability derivative constant terms, which were estimated for a similar quadrotor platform in [55].

The control efforts are determined using a simple feedback control scheme where the desired states are compared to the estimated states:(16)$$\delta_{Lat}=K_{v}\left(y_{des}-\hat{y}\right)$$
(17)$$\delta_{Vert}=K_{w}\left(z_{des}-\hat{z}\right)$$
(18)$$\delta_{Head}=K_{r}\left(\psi_{des}-\hat{\psi }\right)$$
where $K_{v}$, $K_{w}$, and $K_{r}$ are the lateral, vertical, and heading controller gains. A fixed forward speed is used directly as an input to the controller. The closed loop dynamics equations are shown in Equations (Equation 19)–(Equation 21):(19)$$\left[\begin{array}{c} \dot{y}\\ \dot{v} \end{array}\right]=\left[\begin{array}{cc} 0 & 1\\ -Y_{\delta_{Lat}}K_{v} & Y_{v} \end{array}\right]\left[\begin{array}{c} y\\ v \end{array}\right]$$
(20)$$\left[\begin{array}{c} \dot{z}\\ \dot{w} \end{array}\right]=\left[\begin{array}{cc} 0 & 1\\ -Z_{\delta_{Vert}}K_{w} & Z_{w} \end{array}\right]\left[\begin{array}{c} z\\ w \end{array}\right]$$
(21)$$\left[\begin{array}{c} \dot{\psi }\\ \dot{r} \end{array}\right]=\left[\begin{array}{cc} 0 & 1\\ -N_{\delta_{Head}}K_{r} & N_{r} \end{array}\right]\left[\begin{array}{c} \psi \\ r \end{array}\right]$$

Stability of the controlled system can be determined using simple linear stability analysis. In each case, with a selection of $K_{i}>0$ the decoupled linearized systems are stable for small deviations from the equilibrium.

Similar to before, the controller gains can also be moved inside the spatial inner-product:(22)$$u_{i}=\langle \mu ,\sum_{j=1}^{M}K_{i}C_{ij}^{†}F_{j}\rangle$$

Now, the control commands $u_{i}$ can be directly computed from the measured nearness scan using the spatial inner-product. In this case, the second term in the inner-product is a spatially distributed weighting shape that is not only sensitive to the different environment relative states, but is also scaled to produce the desired control commands:(23)$$F_{u_{i}}=\sum_{j=1}^{M}K_{i}C_{ij}^{†}F_{j}$$

For each of the three environment relative states, a different weighting function $F_{u_{i}}$ is used to generate control commands from the measured nearness scans. In this case, the estimated states are linear functions of a single projection, as seen in Equations (Equation 9)–(Equation 11), and so the control command weighting shapes are just scaled versions of the shapes seen in Figure 5.

While a fixed forward speed can work in most environments, performance can be improved by regulating the forward speed. In this case, the speed response from [33] is used for inspiration for the following control law:(24)$$u_{des}=u_{max}(1.0-K_{u_{v}}|\hat{y}|-K_{u_{w}}|\hat{z}|-K_{u_{\psi }}|\hat{\psi }\left|-K_{u_{fwd}}p\left(3\right)\right)$$

Here, the forward speed control gains $K_{u_{v}}$, $K_{u_{w}}$, $K_{u_{\psi }}$, $K_{u_{fwd}}$ are used to tune the forward speed regulation response to errors in the lateral, vertical, and heading states as well as approaching obstacles through the p(3) projection. By regulating the forward speed based on errors in the environment relative states, the system is better equipped to navigate changes in the environment.

## 3. Results

In the following subsections, performance of the algorithm in different simulated environments is presented. In Section 3.1, the performance of the system in a perfect cylinder environment is demonstrated. Zero-mean Gaussian white noise is added to the depth measurements in Section 3.2, demonstrating the algorithm’s robustness to sensor noise. To demonstrate the robustness of the algorithm in nongeneralized environments, results from tests conducted in tunnel and cave environments are presented in Section 3.3.

### 3.1. Performance in Generalized Cylinder Environment

In this section, the algorithm is tested in a simulated perfect cylinder environment to demonstrate the effectiveness of the method. For the first set of tests, the quadrotor system was manually perturbed off the centerline of the cylinder and state estimates generated from the state sensitivity shapes $F_{x_{i}}$ were recorded. No sensor noise was added to any of the tests presented in this subsection. Figure 6, Figure 7 and Figure 8 compare the environment relative state estimates, shown as a red dashed line, to the ground truth, shown in blue. In each of these cases, the estimation error is negligible for small perturbations, however, large perturbations eventually begin to skew the estimated states.

In Figure 9, Figure 10 and Figure 11, the stability of the controller is demonstrated for different perturbations away from the centerline of the cylinder. Isolated tests were conducted for each state where the system is initialized with some nonzero initial condition. Control commands were generated using the control sensitivity shapes $F_{u_{i}}$, and control gains $K_{v}=K_{v}=K_{r}=1$ are used simply to prove stability claims. In each case, the nearness controller drives the system back to the centerline of the cylinder, confirming the stability of the controller. With equal gains, the heading control is seen to have the slowest response, which is to be expected based on the quadrotor model stability derivatives.

### 3.2. Robustness to Noise

To evaluate the algorithm’s robustness to additive Gaussian white noise on the depth measurements, the quadrotor platform was piloted through the cylinder environment for 60 s. During the flight, the system was perturbed away from the cylinder centerline along each environment relative state and the sensor data were recorded. Zero-mean Gaussian noise with varying standard deviations was added to the depth measurements offline, and the standard deviation of the error in the state estimates was computed. The results are shown in Figure 12.

The standard deviation of the state estimates remain extremely small relative to the standard deviation of the added sensor noise. Even with large noise distributions, the algorithm is still able to maintain state estimates that are usable for feedback control.

### 3.3. Performance in Nongeneralized Subterranean Environments

Using different SubT simulation environments, the performance of the algorithm in nongeneralized environments is demonstrated. The control gains listed in the Table in Appendix C were used in both the tunnel and cave tests presented. Figure 13 shows different views of the “Simple Tunnel 03” world, which is composed of sloping ramps, curved bends, vertical shafts, and dead-ends.

The speed of the system and the control commands are shown in Figure 14. The system is able to quickly navigate through all sections of the course in 4 min, covering 506 m of tunnel in the process. The average speed of the system was 2.01 m/s. The first section of the environment is the downward sloping ramp shown in Figure 13b. The vertical centering response is demonstrated in the $u_{w}$ plot in Figure 14 between the 20 and 35 s mark where the system is commanded to descend. Perturbations in the $u_{v}$ commands that correlate with the $u_{r}$ commands are indicative of the system navigating a bend in the tunnel. At the 80-s mark, the system approaches the vertical shaft, as seen by the decrease in the forward speed command. Even though the control algorithm is not explicitly designed to navigate vertical shafts, the system is still able to get through the section and continue exploring.

The SubT virtual cave worlds provide even more diverse terrain for systems to navigate, consisting of large caverns connected by tight passageways. Figure 15a shows a top down view of the DARPA SubT sim world “Simple Cave 02”, and a sample of the inside of the environment is shown in Figure 15b. The cave environments deviate massively from the cylinder model used to approximate the local environment, however the algorithm is still able to produce a centering response that keeps the vehicle exploring. The red line in Figure 15 shows the approximate route taken by the system through the cave environment starting at the green dot near the bottom and proceeding counter-clockwise. During this test, the system covered 1.2 km in just under 15 min, with an average speed of 1.35 m/s. The speed of the system and the control commands as it navigates through environment shown in Figure 15 are shown in Figure 16.

## 4. Discussion

In general, the WFI method works well on spatially distributed depth scans, and sensible state sensitivity shapes are presented in Figure 5. The magnitude of the lateral state weighting shape $F_{\hat{y}}$ is greatest at the sides of the vehicle, which makes sense because these are the measurements that should carry the most weight when estimating $\hat{y}$. Similarly, the vertical state weighting shape $F_{\hat{z}}$ magnitude is largest at the top and bottom poles, and it decreases as they get closer to the x−y plane. The heading state sensitivity shape $F_{\hat{\psi }}$ is slightly more interesting. With opposing positive and negative components, this shape becomes sensitive to changes in the heading of the system.

The state estimates from the perfect cylinder environment demonstrated in Section 3.1 were computed using the full state sensitivity shapes. However, different responses can be achieved if subsets of the full shapes are used. For example, if only the front half of the heading state sensitivity shape $F_{\hat{\psi }}$ is used in the spatial inner-product (i.e., measurements from behind the vehicle are not used), a more reactive steering response is produced. By only processing points that are in front of the vehicle, the system is more responsive to changes in the environment in front of it and can steer accordingly. The results presented in Section 3.3 were achieved using the front half of the $F_{\hat{\psi }}$. The same heading state-estimates can be achieved by recomputing the projection coefficient in the $C^{†}$ matrix in Equation (Equation 6). Other responses, like ground and wall following can be achieved through half shapes as well. With small changes to the feedback controller, a ground-following response can be produced by using only the bottom half of the vertical state sensitivity shape $F_{\hat{z}}$. Similarly, using only the left- or right-hemisphere of the lateral state sensitivity shape $F_{\hat{y}}$ can produce a wall-following response.

The algorithm’s robustness to additive Gaussian sensor noise is demonstrated in Section 3.2. No appreciable change in the standard deviation of the state estimate errors is excited until relatively large deviations in the noise are added to the measurements. Modern LiDAR depth sensors have noise distributions with standard deviations to the order of millimeters to centimeters, which is on the lower end of the data presented in Figure 12. The main takeaway here is that this algorithm enables the use of lower fidelity (and often cheaper) sensors whose noise levels are unacceptable for other applications.

The algorithm is also robust to occlusions, or discontinuities in the measurements, as long as the occlusions are symmetric. Occlusions in pointclouds could result from using multiple, nonoverlapping depth sensors. In this study, occlusions in the depth scan were present due to the presence of the quadrotor body. The quadrotor model body is detected by the onboard LiDAR sensor in the SubT sim, as it is impossible to place a spherical depth sensing model on a quadrotor and not experience some occlusions. These points are obviously not measurements of the environment, and so they should not be processed by the spatial inner-product. In this case, these measurements can simply be thrown away, as long as the corresponding point is removed from every other hemisphere quadrant. This ensures that the resulting state estimates are not skewed in any one direction.

One of the biggest advantages to this solution is the computational efficiency and simplicity. The entire control algorithm can be condensed down into a single equation, Equation (Equation 22), which directly produces controls commands from sensor measurements. Producing a single control command requires 2N floating point operations, where N is the number of measurements, and so only 6N floating point operations are required to generate a set of control commands. Actual implementation of the algorithm requires some additional data conditioning and processing, however the entire operation takes less than 0.01 s of processing time. This means that scans from equivalent LiDAR sensor models could be processed at rates up to 100 Hz. Faster processing of sensor measurements enables faster reactions to changes in the environment, in which case the limiting factor becomes the vehicle dynamics and control response.

While this algorithm is computationally efficient and enables fast flight, it can be susceptible to local minima in the environment such as corners, split paths, or unique environments that produce trajectory loops. In the case where the vehicle is approaching a symmetric split path or corner, the system will not deviate from its current trajectory because the local environment may already be balanced. In this case, the controller may not generate steering commands to avoid a collision. This situation can be thought of as an unstable equilibrium state. In real environments, these scenarios are rare because the relative environment is rarely perfectly symmetric, and even sensor noise can be enough to perturb the system off the unstable equilibrium state and back onto a centering trajectory. As mentioned above, trajectory loops can be excited in certain unique environments. For example, if the system were to fly into a large cavern from a small hallway, it may never exit the cavern and get stuck in a continuous loop flying around the cavern.

## 5. Conclusions

Using bio-inspired WFI methods, high-bandwidth sensor data can be efficiently processed to produce a 3D centering response in subterranean environments. The derivation of the sensitivity functions presented in Section 2.2.2 shows how different weighting functions can be used to extract environment relative information. With proper selection of the feedback controller gains, the algorithm is demonstrated to be stable for a range of perturbations from the equilibrium state in the cylinder environment. By integrating across thousands of sensor measurements, the algorithm is robust to additive white noise and occlusions in the depth scans. The 3D centering response is robust to large deviations away from the generalized local environment presented in Section 2.2.1, as demonstrated in the flight tests presented in Section 3.3.

Work on this framework can continue in several different directions. In Section 2.2.1, the local environment was approximated as an infinite cylinder with some known radius *r*. In this framework, computing controller gains requires some initial value for the expected radius of the environment. If the actual radius of the local environment is significantly larger than the expected radius, the system performance will be sluggish as the control commands are effectively scaled down. Conversely, if the actual radius is much smaller than the expected radius, the control commands could grow too large and make the system unstable. Using principals of robust control theory, a dynamic controller can be designed which provides performance and stability guarantees for a range of perturbations to the expected radius.

As noted in Section 3.3, the system is able to navigate a vertical shaft using this 3D centering approach; however, it required the vehicle to slow to a near stop. This subterranean exploration framework could be expanded by determining new sensitivity functions for vertically oriented infinite cylinders. By feeding back on a different set of pooled projections, forward- and lateral-centering could be achieved while the system ascends or descends the vertical shaft. This requires additional logic for determining when the system should switch between the different centering modes. Deeper analysis of different projection values may provide insight into signals that could be used to facilitate the mode switching.

This work focuses on a spherical LiDAR sensor model that produces the same distribution of points each scan. In other words, the viewing angles of each scan point are constant and known. Novel depth sensing modalities, such as millimeter wave radar, also produce depth scans, however the incoming, unfiltered points are often spurious and have no known spatial distribution. Future work can be done to apply this algorithm to sensors with unknown spatial distributions of their depth scan points.

## Figures and Tables

**Figure 1 sensors-22-00849-f001:**
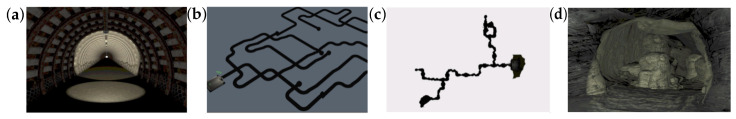
Screenshots of different virtual environments, each spanning hundreds of meters in length and width. (**a**) A simple, straight tunnel environment of radius 1.75 m and length 600 m. (**b**) Overview of a larger, more complex tunnel environment with an average radius of 1.75 m and spanning approximately 200 m width × 300 m length. (**c**) Top-down view of a cave environment spanning approximately 600 m length × 425 m width, with an average cave radius greater than 5 m. (**d**) Entrance to a cave world environment, spanning approximately 10 m width × 15 m height × 20 m length.

**Figure 2 sensors-22-00849-f002:**
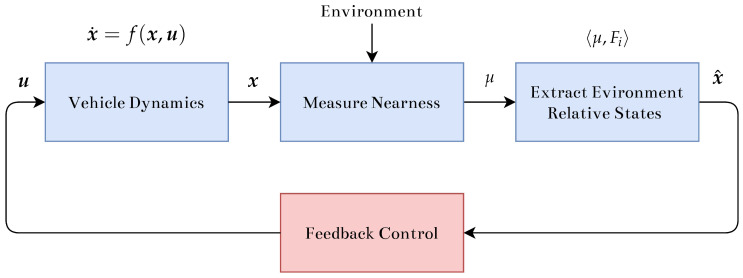
Process flow diagram demonstrating interactions between main subsystems. Nearness is measured from environment and environment relative states are extracted and used for control feedback, ultimately affecting vehicle dynamics.

**Figure 3 sensors-22-00849-f003:**
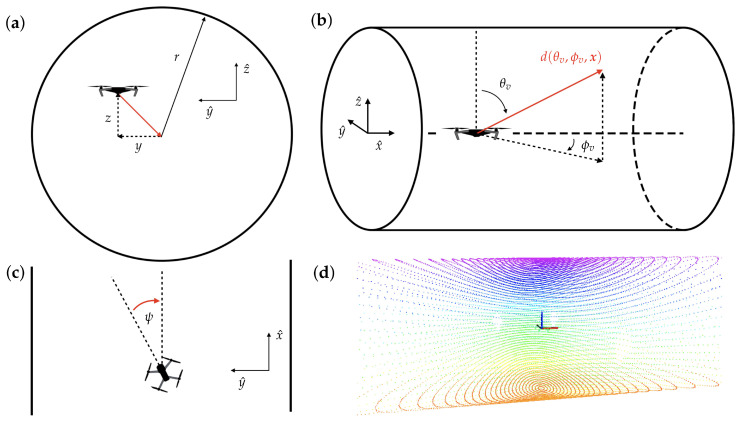
Different views of an infinite cylinder environment are shown with corresponding environment relative variables. (**a**) View looking down cylinder. (**b**) Full 3D view of cylinder environment. (**c**) Top-down view of cylinder. (**d**) Example of a measured depth scan in a cylinder environment using spherical LiDAR sensor. The points are colored based on their *z* position, with warmer colors representing points below the x-y plane and cooler colors representing points above the x-y plane.

**Figure 4 sensors-22-00849-f004:**
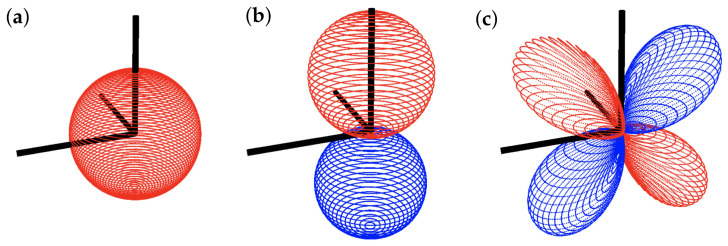
Examples of spherical harmonic shapes, where red points have positive magnitude and blue points have negative magnitude. (**a**) $Y_{0}^{0}$ Mode (**b**) $Y_{0}^{1}$ Mode (**c**) $Y_{-1}^{2}$ Mode.

**Figure 5 sensors-22-00849-f005:**
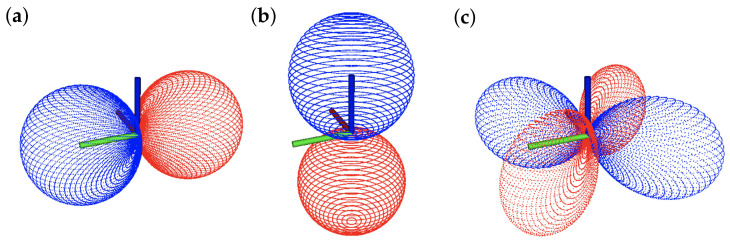
State sensitivity shapes where red points have positive magnitude and blue points have negative magnitude. Origin axis markers are of length 0.5 m. In this linearized example, state sensitivity shapes are simply scaled versions of Laplace spherical harmonics. (**a**) Lateral state *y* sensitivity shape $F_{\hat{y}}$; (**b**) vertical state *z* sensitivity shape $F_{\hat{z}}$; (**c**) heading state ψ sensitivity shape $F_{\hat{\psi }}$.

**Figure 6 sensors-22-00849-f006:**
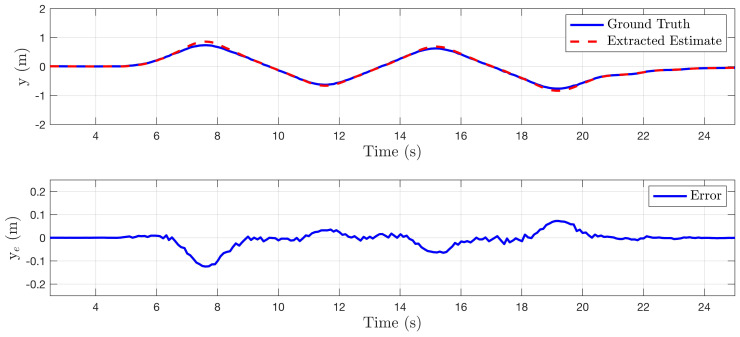
(**top**) Comparison of estimated lateral distance of cylinder centerline to ground truth as system moves back and forth laterally in environment. (**bottom**) State error plotted as a function of time.

**Figure 7 sensors-22-00849-f007:**
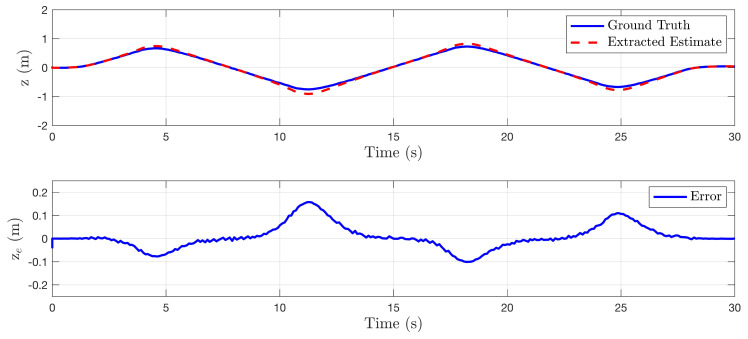
(**top**) Comparison of estimated vertical distance of cylinder centerline to ground truth as system moves up and down in environment. (**bottom**) State error plotted as a function of time.

**Figure 8 sensors-22-00849-f008:**
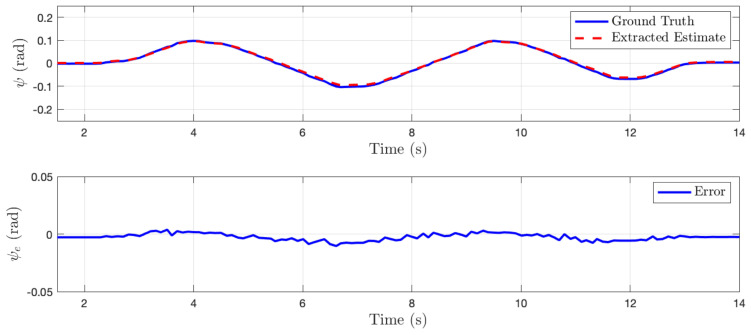
(**top**) Comparison of estimated heading relative to cylinder centerline to ground truth as system yaws back and forth. (**bottom**) State error plotted as a function of time.

**Figure 9 sensors-22-00849-f009:**
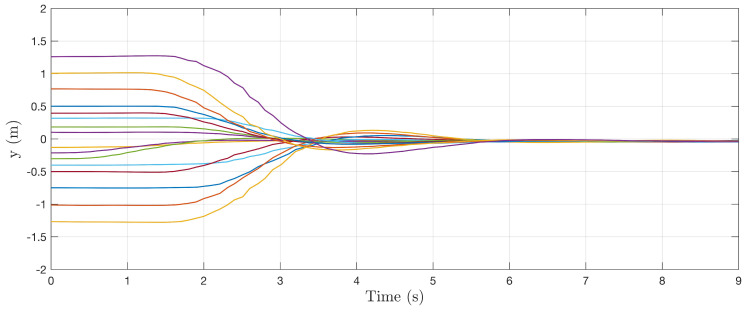
Collection of system responses to a nonzero initial lateral state. Each line represents a different trajectory with a unique nonzero initial perturbation to lateral position.

**Figure 10 sensors-22-00849-f010:**
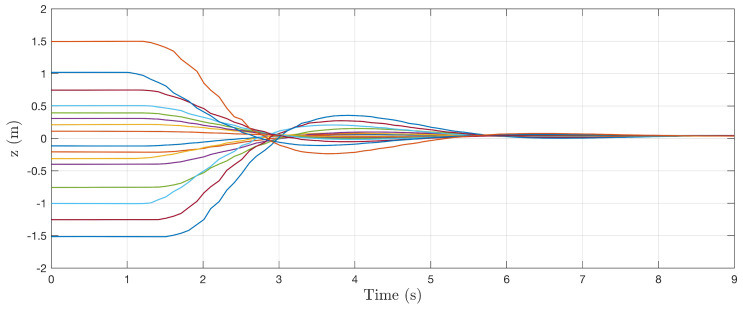
Collection of system responses to a nonzero initial vertical state. Each line represents a different trajectory with a unique nonzero initial perturbation to vertical position.

**Figure 11 sensors-22-00849-f011:**
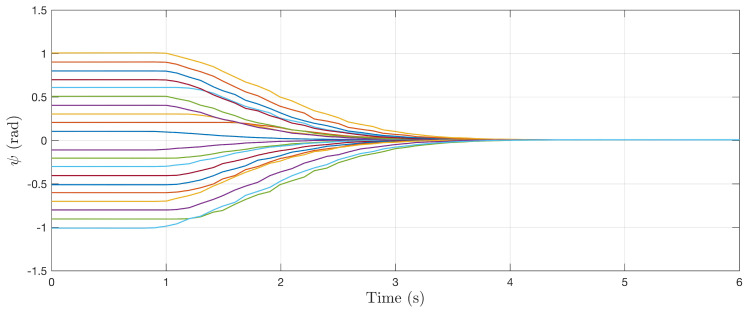
Collection of system responses to a nonzero initial heading state. Each line represents a different trajectory with a unique nonzero initial perturbation to heading of system.

**Figure 12 sensors-22-00849-f012:**
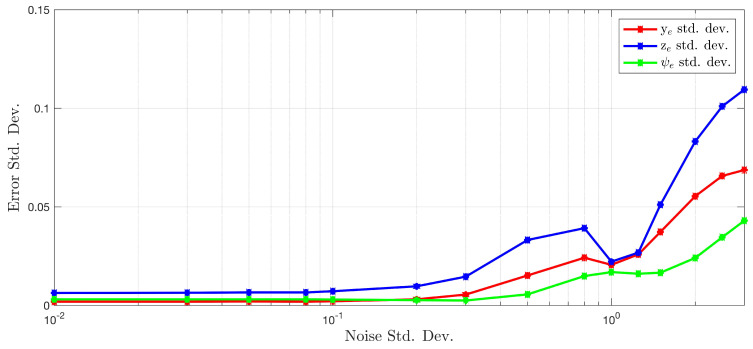
Evolution of measurement error distributions as standard deviation of sensor noise increases.

**Figure 13 sensors-22-00849-f013:**
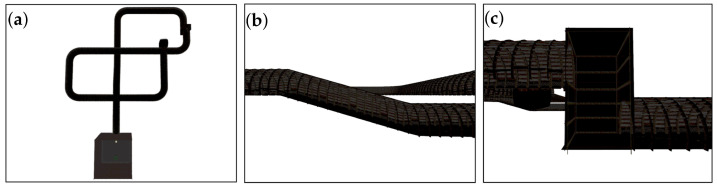
DARPA SubT simulation world: Simple Tunnel 03, 120 m × 100 m × 10 m. (**a**) Top-down view of tunnel environment showing banked turns. (**b**) Side-view of a ramp section where a descent of approximately 7 m occurs over 20 m of forward motion. (**c**) Side-view of vertical shaft section, which spans about 10 m vertically.

**Figure 14 sensors-22-00849-f014:**
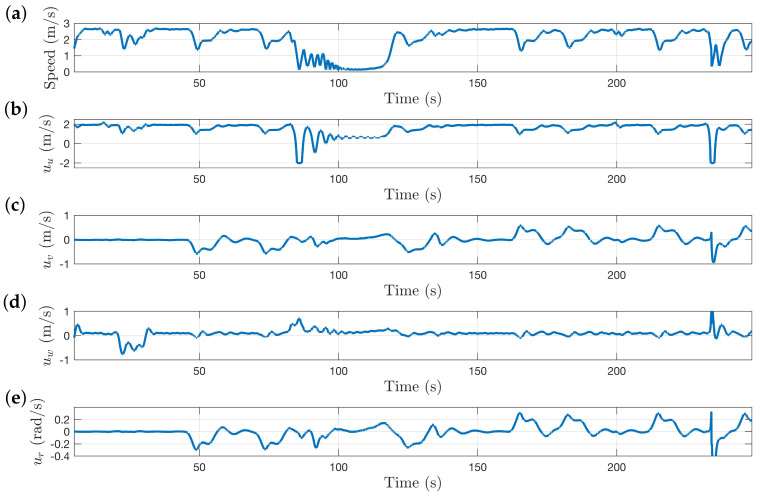
Plots showing speed of system and control commands. (**a**) Speed of system. (**b**) Forward speed command. (**c**) Lateral speed command. (**d**) Vertical speed command. (**e**) Yaw rate command.

**Figure 15 sensors-22-00849-f015:**
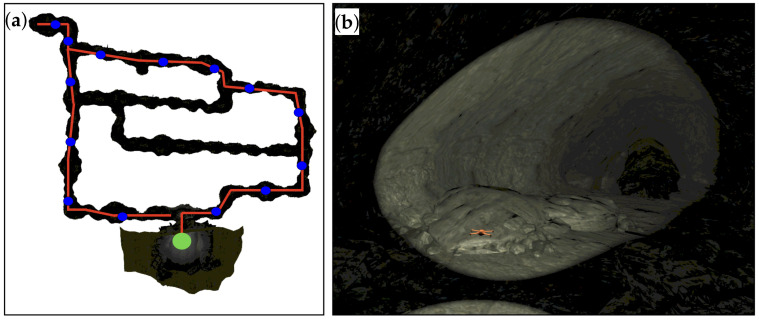
DARPA SubT simulation world: Simple Cave 02, 230 m × 170 m × 75 m (**a**) Top-down view of cave environment consisting of large unstructured tunnels. The red line in Figure 15 shows the approximate route taken by the system through the cave environment starting at the green dot near the bottom and proceeding counter-clockwise. (**b**) View of the orange quadrotor system moving through a larger cavern.

**Figure 16 sensors-22-00849-f016:**
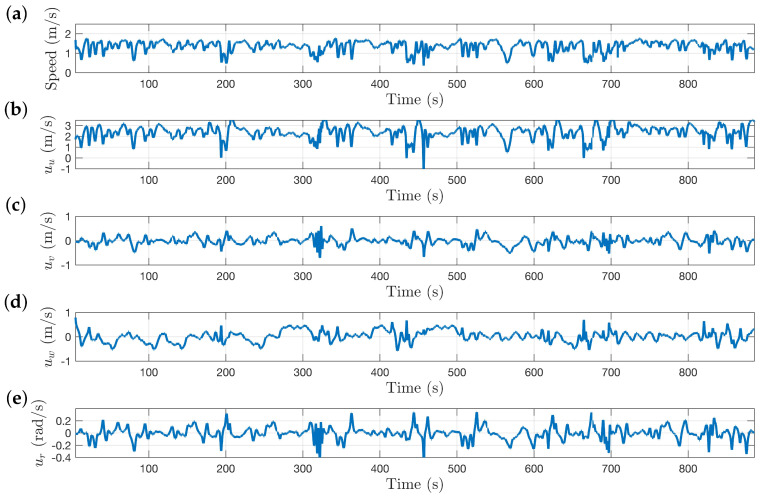
Plots showing speed of system and control commands as it navigates through environment shown in Figure 15. (**a**) Speed of the system. (**b**) Forward speed command. (**c**) Lateral speed command. (**d**) Vertical speed command. (**e**) Yaw rate command.

## Data Availability

Not applicable.

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
