# Peer review of "Lidar-Based Navigation of Subterranean Environments Using Bio-Inspired Wide-Field Integration of Nearness"

_sensors, 2022, doi:10.3390/s22030849_

Round 1
Reviewer 1 Report
Article is interesting, the subject is current and has useful value. In order to enhance the article quality, I suggest the following remarks be taken into account:
- Figure 1 should be moved to Section 2.
- Please add the Figure with marked quantities used in Section 2.1.
- Please give the full name of the acronyms the first time they appear in the text (for example ROS).
- Line 145: ‘Equation 1’ instead of ‘Equation 2’.
- Figure 2 seems to be insufficiently described. Please expand the description.
- Line 193: Incorrect mathematical notation, ‘l ∈ Z, m ∈ Z’ instead of ‘{l, m} ∈ Z’.
- Please mark the matrix\vector in bold (for example line 197).
- The References should be extended to include the publications on intelligent solutions that refer to different types of navigation, for instance:
- Wooden, M. Malchano, K. Blankespoor, A. Howardy, A. A. Rizzi and M. Raibert, "Autonomous navigation for BigDog," 2010 IEEE International Conference on Robotics and Automation, 2010, pp. 4736-4741, doi: 10.1109/ROBOT.2010.5509226.
- Borkowski P. „Computational mathematics in marine navigation” Scientific Journals of the Maritime University of Szczecin no. 21(93), 2010 (20-27)
- A. Bagnell, D. Bradley, D. Silver, B. Sofman and A. Stentz, "Learning for Autonomous Navigation," in IEEE Robotics & Automation Magazine, vol. 17, no. 2, pp. 74-84, June 2010, doi: 10.1109/MRA.2010.936946.
Author Response
Thank you for taking the time to review my manuscript submission. Please see the attached response to the provided comments.

Reviewer 2 Report
The paper presents a bio-inspired wide-field integration method for underground navigation where the main sensing information are the depth measurements. Both the introduction and the number of references is adequate for the correct understanding of the work presented. The work does not stand out for its novelty since this type of control has already been well studied. However, the good results described both in response time and computational efficiency as well as the robustness to occlusions and noise outweigh the proposed work.
The following points should be noted:
Line 23: the acronym for GPS does not appear.
Line 81: misspelling: mutirotor
Line: 89: missing information about LPTC
Equation 1 is not referenced
Line 343: "The" is repeated twice
I miss a section or paragraph of "future work"
Author Response

(The authors gave the same response as above.)

Round 2
Reviewer 1 Report
From my side the work is accepted in this new version.
Author Response
Thank you for your review.